# Association between Survival Duration of Older Patients with Advanced Unresectable Pancreatic Cancer and Appetite Loss: A Retrospective Cohort Study

**DOI:** 10.3390/healthcare10122525

**Published:** 2022-12-14

**Authors:** Ryuichi Ohta, Yoshihiro Moriwaki, Chiaki Sano

**Affiliations:** 1Community Care, Unnan City Hospital, 699-1221 96-1 Iida, Daito-cho, Unnan 699-1221, Japan; 2Department of Surgery, Unnan City Hospital, 699-1221 96-1 Iida, Daito-cho, Unnan 699-1221, Japan; 3Department of Community Medicine Management, Faculty of Medicine, Shimane University, 89-1 Enya cho, Izumo 693-8501, Japan

**Keywords:** unresectable, pancreatic cancer, older patient, rural, appetite loss, mortality

## Abstract

This retrospective cohort study clarified associations between trajectories in palliative care and appetite loss among older patients with advanced unresectable pancreatic cancer and reviewed pancreatic cancer diagnosis among these populations in rural community hospitals. Patients aged >65 years and with pancreatic cancer in a rural community hospital were enrolled. The primary outcome was survival duration from the time of pancreatic cancer diagnosis. Participants were divided into those with and without appetite loss. Cumulative event-free survival rates were calculated using the Kaplan–Meier method, analyzed using the log-rank test, and stratified by factors with statistically significant between-group differences (serum albumin). The mean participant age was 84.14 (SD, 8.34) years; 31.4% were men. Significant between-group differences were noted in albumin concentration and survival duration. Kaplan–Meier curves showed a significant between-group difference in survival probability (*p* < 0.001). Survival duration significantly differed after stratification by albumin level (*p* < 0.001). Appetite loss may be a useful symptom for predicting mortality among older patients with unresectable pancreatic cancer, and hypoalbuminemia may accelerate deterioration in their conditions. Accordingly, subjective appetite loss observed by patients and families should be assessed to predict mortality, and it is advisable for physicians to promptly discuss relevant and advanced directives at appropriate timings.

## 1. Introduction

Diagnosing pancreatic cancer may be difficult because of the various aging-induced symptoms. However, the number of patients with pancreatic cancer is increasing owing to the aging population [1]. Although diagnostic and therapeutic techniques for pancreatic cancer have improved, many pancreatic cancers are still diagnosed in advanced stages and have poor prognoses [2]. Owing to several other symptoms that older people experience, critically alarming pancreatic cancer symptoms remain unnoticed [3,4], which may impact mortality [5]. In particular, the symptoms of pancreatic cancer in older patients are nonspecific, such as fatigue, weight loss, and appetite loss, making early detection difficult [1]. Older patients may experience some of these symptoms regularly, thus failing to seek medical care despite progression of the cancer [6,7].

Moreover, older adults usually have decreased susceptibility to symptoms that can delay the detection of critical diseases, including pancreatic cancer [8]. Cognitive function also declines, which deters perception of symptoms [9]. Transient symptoms may be missed, and only progressing symptoms of advanced cancer may be detected. Therefore, pancreatic cancer are often diagnosed in advanced stages and are already difficult to treat.

Although challenging, clarifying the symptoms and progression of pancreatic cancer among older adults at the time of diagnosis is necessary for early treatment [1,2]. Some studies have suggested no difference in the effect of chemotherapy between older and younger generations [10,11]. However, there is little evidence supporting treatment for older patients aged >80 years with pancreatic cancer since treatment may be less effective in these populations, thus compromising their quality of life [12]. Accordingly, palliative care may be chosen to improve the quality of residual life in these patients.

For palliative care, the prognosis should be discussed with patients and families to prepare them for end-of-life care based on specific symptoms. Based on their prognosis, they can manage their lives and prepare for events and activities [11]. Prognosis can be predicted based on symptoms and biomarkers [12,13]. However, current epidemiological data on pancreatic cancer in these populations, especially those living in remote areas, remain limited. One of the symptoms related to mortality in patients with pancreatic cancer is appetite loss [14,15]. Appetite loss among older people can quickly cause nutritional deficiency such as vitamin deficiency, leading to mortality and morbidity [14,15,16]. Appetite loss can be detected even by lay people based on patient complaints [17]. For a better quality of life among older patients with advanced unresectable pancreatic cancers, the association between detectable symptoms such as appetite loss and mortality should be clarified, hence the aim of the present study. Clarification of their trajectories in palliative care based on appetite can help medical professionals, patients, and their families [18,19]. Their quality of life with cancer should be driven by appetite. Therefore, this study aimed to clarify the association between appetite loss among old patients with advanced unresectable pancreatic cancer in old patients in rural community hospitals.

## 2. Methods

This was a retrospective cohort study in patients aged >65 years who were diagnosed with advanced unresectable pancreatic cancers in a rural community hospital.

### 2.1. Setting

Unnan City is one of the most rural cities in Japan and is located in the southeast of Shimane Prefecture. In 2020, the total population of Unnan was 37,638 (18,145 males and 19,492 females), with 39% individuals aged >65 years, and this is expected to reach 50% by 2025. There are 16 clinics, 12 home care stations, 3 visiting nursing stations, and only 1 public hospital (Unnan City Hospital) in the city [20]. Unnan City Hospital has 281 beds comprising 160 acute and 43 comprehensive care beds, 30 rehabilitation beds, and 48 chronic care beds. There are 14 medical specialties, and the nurse-to-patient ratio is 1:10 for acute care, 1:13 for comprehensive care, 1:15 for rehabilitation, and 1:25 for chronic care [20].

### 2.2. Participants

All patients aged > 65 years and diagnosed with irresectable pancreatic cancer in Unnan City Hospital between 1 April 1 2001 and 31 December 31 2021 were included in this study.

### 2.3. Measurements

#### 2.3.1. Primary Outcome

The primary outcome was survival duration from the time of diagnosis of pancreatic cancer to death. These data were obtained from the electronic medical records at Unnan City Hospital.

#### 2.3.2. Independent Variable

Information regarding appetite loss at diagnosis was collected from the electronic medical records.

#### 2.3.3. Covariates

The following data from the electronic health records of Unnan City Hospital were also collected: age, sex, body mass index (BMI), albumin (g/dL) for the assessment of nutrition and serum creatinine, estimated glomerular filtration rate (eGFR) for the assessment of renal function, hemoglobin level, care level based on the Japanese long-term insurance system [21], Charlson comorbidity index (CCI) based on their past medical histories to assess their severities (the presence of heart failure, myocardial infarction, asthma, chronic obstructive pulmonary diseases, kidney diseases, liver diseases, diabetes mellitus, brain infarction, brain hemorrhage, hemiplegia, connective tissue diseases, dementia, and cancer) [22], and cognitive and motor components; Functional Independence Measure (FIM) total score at admission, which were measured by therapists as an indicator of patients’ ADL; time of diagnosis of pancreatic cancer and death and the presence of chemotherapy; and care level based on the Japanese long-term insurance system (numbered from 1 to 5, with 1 being the least dependent and 5 being severely dependent) [20]. Care level was divided into two groups (dependent: ≥1 and non-dependent: <1) based on the burden on caregivers and families. Appetite loss was assessed based on the Edmonton Symptom Assessment System, and this information was obtained from the review of systems by the nurses at the first visit for each patient [23].

### 2.4. Statistical Analysis

The patients were divided into two groups: with and without appetite loss. Student’s *t*-test was performed on parametric data, and the Mann–Whitney U test was performed on non-parametric data. Cumulative event-free survival rates were calculated using the Kaplan–Meier method, analyzed using the log-rank test, and stratified by factors with statistically significant differences between the groups. Patients with missing data were excluded from the analysis. Regarding the sample size calculation, 30 participants were determined to provide 80% statistical power and 5% type 1 error to detect a significant difference between two survival curves of patients with survival rate of 0.05 and 0.5 in the group with appetite loss and no appetite loss, respectively [15]. Statistical significance was set at *p*-value < 0.05. All statistical analyses were performed using EZR (Saitama Medical Center, Jichi Medical University, Saitama, Japan), a graphical user interface for R (The R Foundation, Vienna, Austria) [24].

### 2.5. Ethical Consideration

The hospital ensured anonymity and confidentiality of the patients’ information. Information related to this study was posted on the hospital website without disclosing any details about the patients. To address any questions regarding this study, the contact information of the hospital representative was listed on the website. The purpose of this study was explained, and informed consent was obtained from all participants. The Clinical Ethics Committee of our institution approved this study (approval code: 20210024).

## 3. Results

### 3.1. Demographic Data of the Participants

Of the 99 patients diagnosed with pancreatic cancer, 90 patients were aged >65 years. After excluding patients who had undergone surgery, 35 participants were evaluated (Figure 1). Overall, 17 participants had appetite loss. The mean age of the participants was 84.14 (SD = 8.34) years, and 31.4% were males. Significant differences were noted in the albumin concentration and survival duration between the groups (Table 1).

### 3.2. Regression Model Results

Kaplan–Meier curves showed a significant difference in the probability of survival between the two groups (*p* < 0.001) (Figure 2A). Following stratification by the presence of hypoalbuminemia (<3 g/dL of serum albumin level), the difference in survival duration between the groups was significant (*p* < 0.001) (Figure 2B).

## 4. Discussion

This retrospective cohort study in a rural context showed that appetite loss is a symptom useful for predicting mortality among older patients with unresectable pancreatic cancer. Hypoalbuminemia may accelerate deterioration in these patients. In diagnosing pancreatic cancer among older patients, appetite loss should be assessed to predict mortality and facilitate effective discussion with patients and families regarding advanced directives and palliative care.

The association between appetite and mortality in patients with pancreatic cancer can be explained by various factors such as inflammation, infections, and hormonal imbalances [25,26]. This study demonstrates a positive association between appetite loss and mortality rate in patients with unresectable pancreatic cancer. Pancreatic cancer induces hormonal imbalance, triggering the hypo- and hypersecretion of insulin [27], which may cause rapid changes in serum glucose, impinging on the consciousness of patients and their appetite conditions. Therefore, hypoglycemia and appetite loss may increase the mortality rates in patients [15,28]. In addition, pancreatic cancer gradually invades the surrounding and distant organs, causing inflammation [25,26]. In this study, the survival duration and appetite loss were measured based on diagnosis. Although the staging of cancer and cognitive and physical functions were not different between groups, the rate of progression of pancreatic cancer may vary owing to differences in the characteristics of malignancy [1]. This study could not investigate the pathological findings of the cancer because of its unresectable nature. Considering the high mortality in the group with appetite loss, appetite loss at the time of diagnosis could be related to enhanced cancer progression among patients with pancreatic cancer.

Most participants in this study were aged >80 years, whose organ systems could have deteriorated because of tissue atrophy [29]. Atrophy of the gastrointestinal mucosa decreases the absorption of vital nutrients, including water-soluble vitamins [29,30]. Moreover, the risk of *Helicobacter pylori* infection is high among older rural patients because of the consumption of unclean drinking water [31]. Infection further atrophies the gastrointestinal tract [32]. Older people in rural settings tend to have reduced absorption of various nutrients such as vitamin B and other minerals, impacting gastric acid and internal factors because of atrophy of the gastric mucosa [33]. Therefore, older patients could become malnourished more quickly than younger patients. In this study, adjusting for hypo-nutrition by measuring serum albumin level showed that appetite loss could affect the mortality of patients with unresectable pancreatic cancer. Moreover, appetite loss may be affected not only by albumin levels but also by malabsorption of other vitamins and minerals.

Advanced directives and palliative care can be efficiently discussed in rural contexts by confirming the presence of appetite loss [34,35]. The study showed that appetite loss consequently limits the survival of patients with unresectable pancreatic cancer. These findings can then guide the patients, families, and medical professionals in rural settings. Most rural settings lack sufficient healthcare resources, and patients and families tend to depend on healthcare professionals in decision making regarding palliative care [36]. The assessment of appetite loss was performed by the nurses listening to patients and families. [37]. Decision making regarding observation and assessment is vital for administrating effective advanced directives [23,38]. Therefore, assessment of appetite loss by patients and their families is useful for advanced directives and palliative care for unresectable pancreatic cancers.

This study has some limitations. First, this study was performed at a single rural community hospital in Japan with small sample size, affecting its generalizability to other populations. Nevertheless, we calculated the sample size with adequate statistical power. Thus, this study can serve as a foundation for investigating subjective symptoms and survival rates among older patients with unresectable pancreatic cancer in rural settings. Future studies should investigate the effect of appetite loss on the survival duration of these patients in other countries. Second, the exclusion of patients who underwent surgery for pancreatic cancer may have affected the results. As cancer staging was performed properly by surgeons, there might be patients with advanced cancer with the same survival rate. Future studies should investigate patients with pancreatic cancer at various stages of appetite loss and survival durations. Furthermore, selection bias could not be avoided in this study. Most intensive care procedures, such as surgery and chemotherapy, are performed in urban general hospitals. The results of this study should be used in rural community medicine as advanced directives for older rural people.

## 5. Conclusions

Appetite loss may be a useful symptom for predicting mortality among old patients with unresectable pancreatic cancer. Hypoalbuminemia may accelerate deterioration in these patients. In diagnosing pancreatic cancer among older patients, the presence of subjective appetite loss, which is observed by patients and families, should be assessed to predict mortality. The information of loss of appetite can be used as one of the vital information to promote prompt discussion with patients and families regarding advanced directives at appropriate times.

## Figures and Tables

**Figure 1 healthcare-10-02525-f001:**
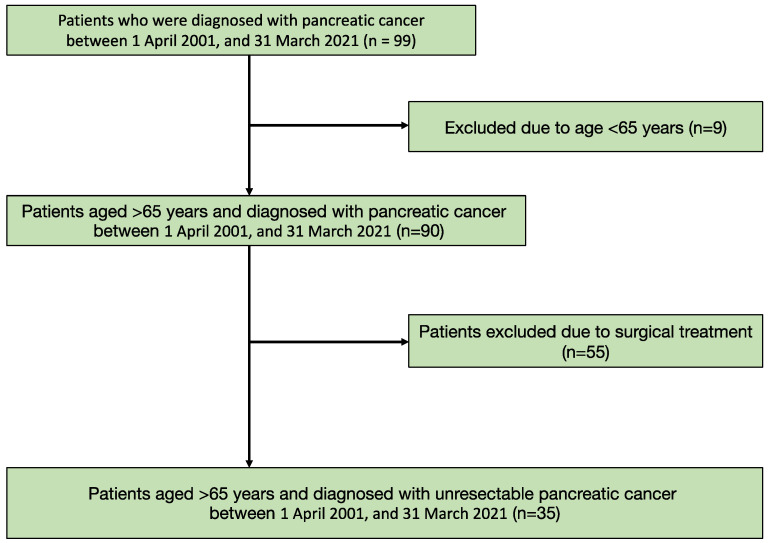
Flow chart of the patients’ selection.

**Figure 2 healthcare-10-02525-f002:**
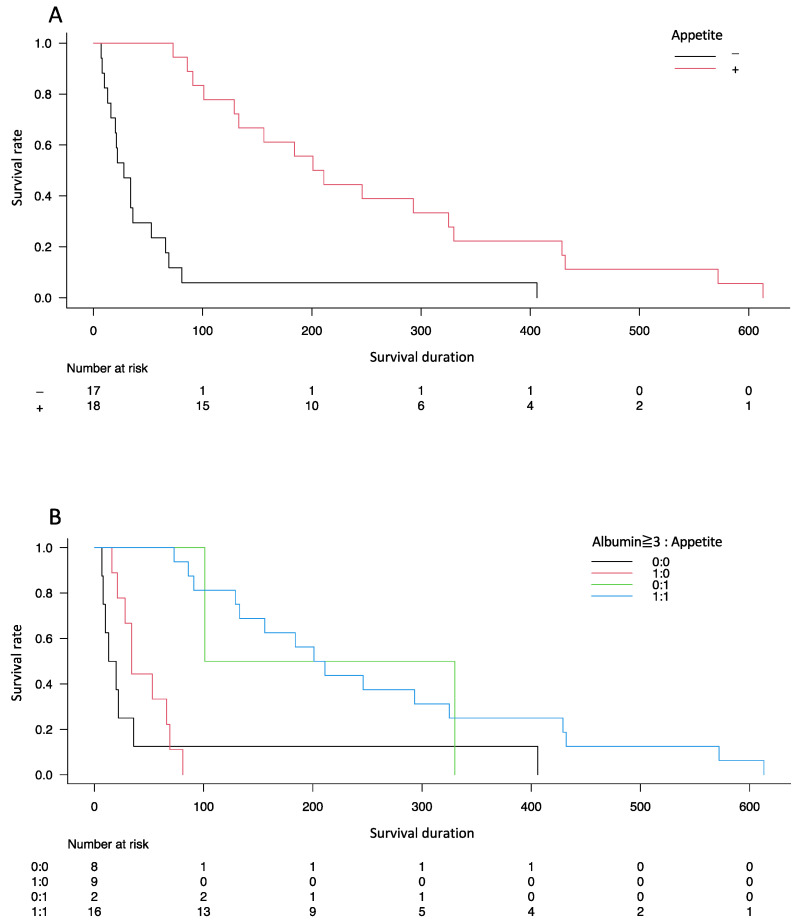
Kaplan–Meier curves showing the probability of survival from the time of diagnosis in patients diagnosed with unresectable pancreatic cancer. Difference in the probability of survival between the two groups (**A**) with stratification by the presence of hypoalbuminemia (<3 g/dL of serum albumin level) (**B**).

**Table 1 healthcare-10-02525-t001:** Demographic data of the participants.

	Appetite			
Factor	Total	Yes	No	*p*-Value
N	35	18	17	
Age, mean (SD)	84.14 (8.34)	82.50 (8.81)	85.88 (7.68)	0.236
Male sex (%)	11 (31.4)	4 (22.2)	7 (41.2)	0.289
Albumin, mean (SD)	3.23 (0.73)	3.56 (0.51)	2.87 (0.78)	0.004
Height, mean (SD)	149.02 (11.65)	148.07 (12.85)	150.02 (10.54)	0.629
Body weight, mean (SD)	43.71 (10.18)	41.43 (9.05)	46.12 (11.02)	0.178
BMI, mean (SD)	19.57 (3.06)	18.90 (3.20)	20.28 (2.83)	0.186
eGFR	68.74 (20.35)	73.92 (17.37)	63.26 (22.31)	0.123
Hemoglobin, mean (SD)	10.81 (2.07)	10.94 (2.24)	10.68 (1.93)	0.713
CA19-9, median (IQR)	187.60 (2, 4,969,129.30)	102.95 (2, 169,009.50)	258.40 (2, 4,969,129.30)	0.594
FIM score at diagnosis				
Total FIM score, mean (SD)	89.89 (39.82)	97.00 (32.70)	82.35 (46.01)	0.283
Motor domain score, mean (SD)	63.91 (31.48)	68.11 (27.86)	59.47 (35.22)	0.425
Cognitive domain score, mean (SD)	27.97 (11.42)	30.83 (8.46)	24.94 (13.50)	0.129
Usage of chemotherapy (%)	6 (17.1)	2 (11.1)	4 (26.7)	0.420
Survival duration, median (IQR)	86 (7, 613)	206 (73, 613)	28 (7, 406)	<0.001
CCI (%)				
2	1 (2.9)	1 (5.6)	0 (0.0)	0.219
3	2 (5.7)	1 (5.6)	1 (5.9)	
4	17 (48.6)	11 (61.1)	6 (35.3)	
5	7 (20.0)	1 (5.6)	6 (35.3)	
6	5 (14.3)	2 (11.1)	3 (17.6)	
7	3 (8.6)	2 (11.1)	1 (5.9)	
Heart failure (%)	1 (2.9)	0 (0.0)	1 (5.9)	0.486
MI (%)	2 (5.7)	1 (5.6)	1 (5.9)	1
Peptic ulcer (%)	4 (11.4)	2 (11.1)	2 (11.8)	1
Kidney disease (%)	1 (2.9)	1 (5.6)	0 (0.0)	1
Liver disease (%)	2 (5.7)	1 (5.6)	1 (5.9)	1
DM (%)	9 (25.7)	4 (22.2)	5 (29.4)	0.711
Brain infarction (%)	5 (14.3)	2 (11.1)	3 (17.6)	0.658
Brain hemorrhage (%)	1 (2.9)	1 (5.6)	0 (0.0)	1
Connective tissue disease (%)	1 (2.9)	0 (0.0)	1 (5.9)	0.486
Dementia (%)	4 (11.4)	1 (5.6)	3 (17.6)	0.338
Cancer (%)	6 (17.1)	3 (16.7)	3 (17.6)	1
Dependent condition (%)	14 (40.0)	7 (38.9)	7 (41.2)	1

Abbreviation: BMI, body mass index; FIM, Functional Independence Measure; CCI, Charlson comorbidity index; MI, myocardial infarction; DM, diabetes mellitus.

## Data Availability

The datasets used and/or analyzed during the current study may be obtained from the corresponding author upon reasonable request.

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
