# Peer review of "Association between Survival Duration of Older Patients with Advanced Unresectable Pancreatic Cancer and Appetite Loss: A Retrospective Cohort Study"

_healthcare, 2022, doi:10.3390/healthcare10122525_

Round 1
Reviewer 1 Report
Ohta et al, in their manuscript entitled “Relationship between Survival Duration of Older Patients with Advanced Unresectable Pancreatic Cancer and Appetite Loss: A Retrospective Cohort Study “describes a retrospective cohort study in a rural community of Japan. People enrolled (99) are patients older than 65 years with unresectable pancreatic cancer. The Primary outcome is survival duration from time of diagnosis to cancer death. Appetite loss and hypoalbuminemia are considered by authors as negative prognostic factors of survival and can be assessed to predict mortality.
The manuscript is interesting. It si well written . I recommend for the publication after minor revision.
Minor points:
Do you have reported in your retrospective cohort some other cachexia parameters (i.e mass loss) that can be correlate to the grade of appetite loss and survival ?
Author Response
Responses to the reviewer’s comments
Thank you for reviewing our manuscript and providing suggestions for its improvement. We have revised all the contents based on the reviewer’s comments. Our revisions are indicated in red font here and in the manuscript. We hope that the revised manuscript meets the journal’s requirements and can now be considered for publication.
Ohta et al, in their manuscript entitled “Relationship between Survival Duration of Older Patients with Advanced Unresectable Pancreatic Cancer and Appetite Loss: A Retrospective Cohort Study “describes a retrospective cohort study in a rural community of Japan. People enrolled (99) are patients older than 65 years with unresectable pancreatic cancer. The Primary outcome is survival duration from time of diagnosis to cancer death. Appetite loss and hypoalbuminemia are considered by authors as negative prognostic factors of survival and can be assessed to predict mortality.
The manuscript is interesting. It is well written. I recommend for the publication after minor revision.
Minor points:
Do you have reported in your retrospective cohort some other cachexia parameters (i.e mass loss) that can be correlate to the grade of appetite loss and survival?
Response:
Thank you for the valuable feedback. Unfortunately, we did not assess the degree of appetite loss and solely assessed the presence of appetite loss.
Reviewer 2 Report
Anyone can relatively assess the appetite loss, still some definition should be considered. The 35 patients were very old and the assessment of the appetite is difficult and also, appetite loss is parallel with advancing in age and disease progression.
The sample size is very small even if the authors mentioned that their study achieved statistical power and 5% type 1 error.
The only significant difference between the two groups was related to survival which is also related to the stage of disease and comorbidities. The significant difference for albumin demonstrates the appetite loss.
Poor prognosis in inoperable pancreatic cancer is well known and the question is by how much survival would be improved if considering nutritional support by using enteral and parenteral methods in order to substitute the appetite loss.
Therefore, the conclusion that effective discussion with patients and families regarding advanced directives is unclear.
Author Response
Responses to the reviewer’s comments
Thank you for reviewing our manuscript and providing suggestions for its improvement. We have revised all the contents based on the reviewers’ comments. Our revisions are indicated in red font here and in the manuscript. We hope that the revised manuscript meets the journal’s requirements and can now be considered for publication.
Anyone can relatively assess the appetite loss, still some definition should be considered. The 35 patients were very old and the assessment of the appetite is difficult and also, appetite loss is parallel with advancing in age and disease progression.
The sample size is very small even if the authors mentioned that their study achieved statistical power and 5% type 1 error.
Response:
Thank you for the insightful feedback. We agree with the comment of the reviewer. The sample size was small, and thus becomes alimitation of our study.
“This study has some limitations. First, this study was performed at a single rural community hospital in Japan with small sample size, affecting its generalizability to other population. Nevertheless, we have calculated the sample size with adequate statistical power. Thus, this study can serve as a foundation for investigating subjective symptoms and survival rates among older patients with unresectable pancreatic cancer in rural settings. Future studies should investigate the effect of appetite loss on the survival duration of these patients in other countries.” (Lines 207–213)
The only significant difference between the two groups was related to survival which is also related to the stage of disease and comorbidities. The significant difference for albumin demonstrates the appetite loss.
Response:
Thank you for the valuable feedback. We consider that this difference can be an avenue for our future study.
Poor prognosis in inoperable pancreatic cancer is well known and the question is by how much survival would be improved if considering nutritional support by using enteral and parenteral methods in order to substitute the appetite loss.
Response:
Thank you for the useful feedback. We consider that even with nutritional support, palliative patients can live longer with their families. The investigation can be the next step for our research.
Therefore, the conclusion that effective discussion with patients and families regarding advanced directives is unclear.
Response:
Thank you for the insightful feedback. We agree with the suggestion of the reviewer. We have revised the corresponding conclusion as follows:
"Accordingly, subjective appetite loss observed by patients and families should be assessed to predict mortality and it is advisable for physicians to promptly discuss relevant and advanced directives at appropriate timings.” (Lines 27–29)
“Appetite loss may be a useful symptom for predicting mortality among old patients with unresectable pancreatic cancer. Hypoalbuminemia may accelerate deterioration in these patients. In diagnosing pancreatic cancer among older patients, the presence of subjective appetite loss which is observed by patients and families should be assessed to predict mortality. The information of loss of appetite can be used as one of the vital information to promote prompt discussion with patients and families regarding advanced directives at appropriate timings.” (Lines 223–228)
Reviewer 3 Report
Authors desribed an important clinical issue. However, the topis is no characterized by novelity. In my opinion, the work have some limitations.
1) In my opinion word "relationship" should be replaced word "association".
2) In the line 34, I do not agree with a continuation of the sentence "early diagnosis". If the diagnosis would be early, the pancreatic cancer would not be a significant problem for clinicians. Interestingly, your next sentence shows this problem. Please change fragment "early diagnosis" in the sentence in the line 34.
3) In my opinion, in the line 39, jaundice is not needed. A jaundice is more specific symptom for the pancreatic cancer compared to wiegt loss or fatigue.
4) The study group is really small and derives from one hospital.
5) Your work did not reveal detailed mechanisms presented dependence. It decreases the quality of article.
6) Why did not you include the patients who had surgery?
7) In my opinion, you made a selection mistake according to the study group.
8) In my opinion, your conclusions are too general and expected before study.
additional comments: The topic is not novel. The work did not reveal detailed mechanisms of dependence. The work has a lot of limitations that were indicted even by authors.
Author Response
Responses to the reviewers’ comments
Thank you for reviewing our manuscript and providing suggestions for its improvement. We have revised all the contents based on the reviewer’s comments. Our revisions are indicated in red font here and in the manuscript. We hope that the revised manuscript meets the journal’s requirements and can now be considered for publication.
Authors described an important clinical issue. However, the topis is no characterized by novelty. In my opinion, the work has some limitations.
- In my opinion word "relationship" should be replaced word "association".
Response:
Thank you for the helpful feedback. We agree with the suggestion of the reviewer. Accordingly, we have changed the term “relationship” to “association.”
- In the line 34, I do not agree with a continuation of the sentence "early diagnosis". If the diagnosis would be early, the pancreatic cancer would not be a significant problem for clinicians. Interestingly, your next sentence shows this problem. Please change fragment "early diagnosis" in the sentence in the line 34.
Response:
Thank you for the valuable feedback. We agree with the suggestion of the reviewer. I have deleted “early diagnosis” in the revised version of the manuscript.
- In my opinion, in the line 39, jaundice is not needed. A jaundice is more specific symptom for the pancreatic cancer compared to wiegt loss or fatigue.
Response:
Thank you for the useful feedback. We agree with the suggestion of the reviewer. I have deleted the term “jaundice” in the revisedmanuscript.
- The study group is really small and derives from one hospital.
Response:
Thank you for the valuable feedback. We agree that the sample size was small. We have added this as a limitation of the study.
“This study has some limitations. First, this study was performed at a single rural community hospital in Japan with small sample size, affecting its generalizability to other population. Nevertheless, we have calculated the sample size with adequate statistical power. Thus, this study can serve as a foundation for investigating subjective symptoms and survival rates among older patients with unresectable pancreatic cancer in rural settings. Future studies should investigate the effect of appetite loss on the survival duration of these patients in other countries.” (Lines 207–213)
- Your work did not reveal detailed mechanisms presented dependence. It decreases the quality of article.
Response:
Thank you for the useful feedback. We agree with the suggestion of the reviewer. We have added the definition of dependent conditions in the method section as follows.
“care level based on the Japanese long-term insurance system (numbered from 1 to 5, with 1 being the least dependent and 5 being severely dependent) [20]. care level was divided into two groups (dependent: ≥1 and non-dependent: <1) based on the burden on caregivers and families.”(Lines 112–115)
- Why did not you include the patients who had surgery?
Response:
Thank you for the useful feedback. We agree with the suggestion of the reviewer. Our research focused on advanced unresectable pancreatic cancer. We described the issue in figure 1 and the limitation part as well as follows.
“Second, the exclusion of patients who underwent surgery for pancreatic cancer may have affected the results. As cancer staging was performed properly by surgeons, there might be patients with advanced cancer with the same survival rate.” (Line 217-224)
- In my opinion, you made a selection mistake according to the study group.
Response:
Thank you for the useful feedback. We agree with the suggestion of the reviewer. Our research focused on advanced unresectable pancreatic cancer. We made selection flow as figure 1 to show clearly our selection flow.
- In my opinion, your conclusions are too general and expected before study.
Response:
Thank you for the insightful feedback. We agree with the suggestion of the reviewer, and have revised the conclusion as follows:
"Accordingly, subjective appetite loss observed by patients and families should be assessed to predict mortality and it is advisable for physicians to promptly discuss relevant and advanced directives at appropriate timings.” (Lines 27–29)
“Appetite loss may be a useful symptom for predicting mortality among old patients with unresectable pancreatic cancer. Hypoalbuminemia may accelerate deterioration in these patients. In diagnosing pancreatic cancer among older patients, the presence of subjective appetite loss which is observed by patients and families should be assessed to predict mortality. The information of loss of appetite can be used as one of the vital information to promote prompt discussion with patients and families regarding advanced directives at appropriate timings.” (Lines 223–228)
- additional comments: The topic is not novel. The work did not reveal detailed mechanisms of dependence. The work has a lot of limitations that were indicted even by authors.
Response:
Thank you for the valuable feedback. We agree with the suggestion of the reviewer. The study has several limitations, and thus careful interpretation if the findings is warranted. We have revised the limitation section as follows:
“This study has some limitations. First, this study was performed at a single rural community hospital in Japan with small sample size, affecting its generalizability to other population. Nevertheless, we calculated the sample size with adequate statistical power. Thus, this study can serve as a foundation for investigating subjective symptoms and survival rates among older patients with unresectable pancreatic cancer in rural settings. Future studies should investigate the effect of appetite loss on the survival duration of these patients in other countries. Second, the exclusion of patients who underwent surgery for pancreatic cancer may have affected the results. As cancer staging was performed properly by surgeons, there might be patients with advanced cancer with the same survival rate. Future studies should investigate patients with pancreatic cancer at various stages of appetite loss and survival durations. Furthermore, selection bias could not be avoided in this study. Most intensive care procedures, such as surgery and chemotherapy, are performed in urban general hospitals. The results of this study should be used in rural community medicine as advanced directives for older rural people.”(Lines 207–220)
Round 2
Reviewer 3 Report
Authors answered to all questions. However, the main limitations are still present. I leave the decision to Editor.